# ArxEval: Evaluating Retrieval and Generation in Language Models for Scientific Literature

## Abstract

Language Models [LMs] are now playing an increasingly large role in information generation and synthesis; the representation of scientific knowledge in these systems needs to be highly accurate. A prime challenge is hallucination; that is, generating apparently plausible but actually false information, including invented citations and nonexistent research papers. This kind of inaccuracy is dangerous in all the domains that require high levels of factual correctness, such as academia and education. This work presents a pipeline for evaluating the frequency with which language models hallucinate in generating responses in the scientific literature. We propose **ArxEval**, an evaluation pipeline with two tasks using ArXiv as a repository: **Jumbled Titles** and **Mixed Titles**. Our evaluation includes fifteen widely used language models and provides comparative insights into their reliability in handling scientific literature.

## 1 Introduction

Large Language Models (LLMs) have emerged as pivotal tools in information access and generation, particularly through their capabilities of producing factually accurate texts. As these models become increasingly integrated into various applications, ensuring the accuracy of their responses has become very important. The performance and reliability of LLMs in generating accurate information are significantly influenced by multiple factors, including training data quality, model architecture design, and post-training optimization processes Naveed et al. (2024), Minaee et al. (2024), Guo et al. (2023).

However, a significant challenge in the deployment of LLMs lies in their propensity to generate nonfactual responses, a phenomenon commonly referred to as hallucination. These hallucinations fundamentally undermine the reliability and faithfulness of LLMs, presenting substantial obstacles to their widespread adoption across various domains Huang et al. (2024), Sahoo et al. (2024). The mitigation of hallucinations has consequently emerged as a critical area of research within the field. While various strategies have been proposed and implemented to reduce hallucinations, showing promising improvements in the faithfulness of LLMs for general-purpose tasks, domain-specific applications remain particularly challenging Tonmoy et al. (2024), Rawte et al. (2023), Berberette et al. (2024).

In this paper, we present a comprehensive study evaluating the extent of hallucination in LLMs under domain-specific prompting, with a particular focus on scientific literature. We develop and implement a systematic evaluation pipeline to assess fifteen prominent open-source LLMs: Qwen 2.5 Yang et al. (2024), Gemma 2 Team et al. (2024), Llama 3 Grattafiori et al. (2024), Phi 3 Abdin et al. (2024), Orca 2 Mitra et al. (2023), Mistral v-0.3 [Team (2024), Deepseek-llm DeepSeek-AI et al. (2024), Olmo-2 OLMo et al. (2024), Mistral-Nemo Team, Eurus-2 Yuan et al. (2024), and Solar-Pro upstage (2024). Our evaluation utilizes the ArXiv dataset Clement et al. (2019) as the primary source of scientific articles, providing a robust foundation for assessing model performance in academic contexts.

The evaluation pipeline **ArxEval** introduces two novel tasks: **Jumbled Titles** and **Mixed Titles**. These tasks are specifically designed to assess the faithfulness of LLMs in retrieving and reasoning about scientific articles under challenging conditions. The models are presented with either jumbled or mixed titles and evaluated not only on their prompt adherence but also on the quality and accuracy of their generated

outputs. By adopting an open-ended evaluation approach, we aim to provide comprehensive insights into the models' capabilities in processing and responding to ambiguous or altered inputs within a domain-specific context, particularly focusing on their ability to maintain factual accuracy while handling complex scientific information.

This study contributes to the growing body of research on LLM reliability and provides valuable insights into the current limitations and capabilities of state-of-the-art language models in handling domain-specific tasks. Our findings have important implications for the development and deployment of LLMs in scientific and academic applications, where maintaining factual accuracy is crucial.

## 2 Related Work

### 2.1 Hallucinations in Large Language Models (LLMs)

Hallucinations in LLMs have been extensively studied and documented. While significant advancements have been made in improving their accuracy and reliability, LLMs have been shown to hallucinate even when tasked with generating responses based on known facts Jiang et al. (2024). Such behavior suggests an inherent limitation in these models, reinforcing the hypothesis that hallucination may be an intrinsic characteristic Banerjee et al. (2024).

### 2.2 Hallucinations in Domain-Specific Settings

#### 2.2.1 Definition and Challenges

Domain-specific hallucinations manifest when LLMs generate inaccurate or fabricated information in specialized fields. In domains like biomedicine, such hallucinations can have serious implications, potentially leading to incorrect medical advice or misinterpretation of research data. The fundamental challenge lies in maintaining factual accuracy while preserving the model's ability to generate coherent and contextually relevant responses.

#### 2.2.2 Causes and Perspectives

Domain-specific hallucinations primarily stem from two factors: deficiencies in training data and limitations in model architecture Dziri et al. (2022). However, recent research presents an alternative viewpoint, suggesting that under certain conditions, hallucinations could be leveraged as a resource for novel problem-solving approaches Sui et al. (2024).

#### 2.2.3 Detection and Evaluation Frameworks

- **DelucionQA** Sadat et al. (2023): A specialized dataset designed for detecting hallucinations in domain-specific question-answering tasks, providing evaluation metrics for retrieval-augmented LLMs.

- **DAHL** Seo et al. (2024): A comprehensive benchmark for evaluating hallucinations in biomedical text generation, featuring atomic unit decomposition and the DAHL Score metric.

### 2.3 Hallucinations in Multimodal Settings

#### 2.3.1 Definition and Challenges

Multimodal hallucinations occur when models generate outputs inconsistent with visual or auditory inputs. This phenomenon is particularly critical in applications like video understanding, where temporal and spatial accuracy are essential Bai et al. (2024).

### 2.3.2 Evaluation Frameworks

- **VidHalluc** Li et al. (2024): A specialized benchmark for evaluating temporal hallucinations in video understanding, assessing multiple dimensions including action recognition and scene transitions.

- **MHaluBench** Chen et al. (2024): A meta-evaluation framework for comprehensive multimodal hallucination detection across diverse categories.

## 2.4 Hallucinations in Natural Language Generation

In natural language generation tasks such as dialogue generation, abstractive summarization, and neural machine translation, hallucinations frequently manifest as plausible but factually incorrect outputs Ji et al. (2023). These inaccuracies significantly impact the reliability and trustworthiness of these models.

## 2.5 Hallucinations in Academic Reference Generation

Academic reference generation represents a critical challenge, with studies demonstrating that even state-of-the-art models frequently generate fabricated or inaccurate citations Agrawal et al. (2024). This limitation underscores the urgent need for continued research in hallucination mitigation, particularly in tasks where factual accuracy is paramount. In addressing these challenges, our work specifically focuses on *Domain-Specific Biases* by leveraging over 150 categories of papers from the ArXiv repository, providing a comprehensive evaluation across diverse academic domains.

# 3 Dataset

In this section, we describe the dataset used to evaluate our two tasks: the **Jumbled Title** task and the **Mixed Title** task. The dataset is derived from the ArXiv repository and organized into 176 categories referring to the subject areas of the papers within the ArXiv dataset, such as Computer Science, Physics, Economics etc. For each category, 3 paper titles are selected, resulting in a total of 528 titles.

Figure 1 illustrates the distribution of titles across subjects. For instance, Computer Science comprises 65 categories (195 titles), whereas Economics is represented by only 3 categories (9 titles).

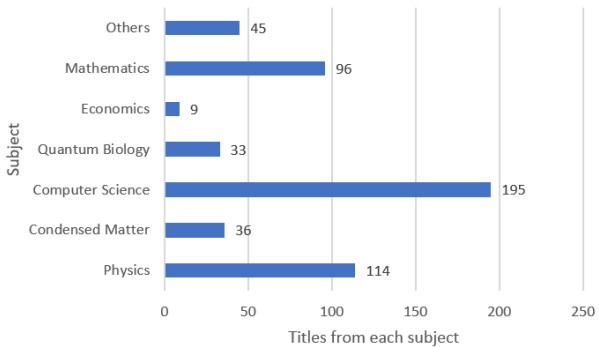

Figure 1: Number of titles from each subject.

Table 1 summarizes the dataset statistics along with readability metrics. The Flesch reading ease scores indicate that the Jumbled Titles fall within the `Very difficult to read. Best understood by university graduates` range, while the Mixed Titles are classified as `Extremely difficult to read. Best understood by university graduates`. Similarly, the Gunning fog index places the Jumbled Titles at a `College graduate` level (score 17) and the Mixed Titles also at a `College graduate` level (score 19). The title lengths vary widely (Jumbled Titles: 2–24 words; Mixed Titles: 8–33 words), ensuring a robust evaluation across diverse input complexities.

| Task | Total | Categories Count | Avg. Len | Range | Readability | |
|---|---|---|---|---|---|---|
| | | | | | Gunning Fog | Flesch |
| Jumbled Titles | 528 | 176(3) | 9.45 | 2–24 | 17 | 16 |
| Mixed Titles | 265 | 176(3) | 18.89 | 8–33 | 20 | 8 |

Table 1: Dataset Statistics for the Jumbled and Mixed Titles Tasks.

### 3.1 Jumbled Title Task

In real-world academic and research settings, users often recall only fragments of paper titles or misremember their exact phrasing. They may reorder words, conflate multiple concepts, or substitute synonymous terms when searching for relevant literature. The **Jumbled Titles** task is designed to reflect this intrinsic difficulty by presenting models with scrambled versions of real paper titles. This approach mimics the challenges of information retrieval, where users provide imprecise search queries due to memory limitations, cognitive biases, or incomplete knowledge.

A robust language model should be able to process these disordered inputs effectively, retrieving relevant research despite the inconsistencies. By evaluating models on their ability to reconstruct meaningful associations from jumbled titles, we assess their resilience in real-world search conditions. This task not only tests a model's capacity to recognize and reassemble key concepts but also highlights its practical utility in assisting researchers who struggle with recalling precise paper titles.

Each title from the original dataset is internally scrambled to produce a jumbled version. Table 2 presents examples of jumbled titles alongside their corresponding original titles.

| Jumbled Title | Original Title |
|---|---|
| Hydrodynamic bubble to obstruction expansion | Hydrodynamic obstruction to bubble expansion |
| with Warm Microwave Background Constraining Inflation Cosmic the | Constraining Warm Inflation with the Cosmic Microwave Background |
| QCD Hadron Colliders Three-Jet Corrections Production Two-Loop at for Leading-Color | Leading-Color Two-Loop QCD Corrections for Three-Jet Production at Hadron Colliders |
| enumeration theorem polynomials Order Pólya's and | Order polynomials and Pólya's enumeration theorem |

Table 2: Examples from the dataset used for the Jumbled Title task.

### 3.2 Mixed Title Task

Scientific discovery often emerges from the intersection of multiple disciplines, where researchers seek to explore new ideas by combining concepts from different fields. For instance, a scientist might ask whether there are existing studies on the integration of quantum mechanics with financial modeling or the application of machine learning in archaeology. Such inquiries reflect the growing importance of interdisciplinary research. Researchers often explore new areas by searching for existing papers that address these cross-disciplinary topics or by proposing novel combinations of ideas.

The **Mixed Titles** task captures this trend by merging two disparate paper titles, assessing whether models can identify relevant papers that address both topics. This task evaluates not only the truthfulness of model-recommended references but also their capacity to facilitate interdisciplinary research. By testing a model's ability to recognize meaningful connections across domains, the task provides insight into how well AI can support scientific innovation and knowledge synthesis.

Scientific discovery often emerges from the intersection of multiple disciplines, where researchers seek to explore new ideas by combining concepts from different fields. For instance, a scientist might ask whether there are existing studies on the integration of *quantum mechanics with financial modeling* or *the application of machine learning in archaeology.* Such inquiries reflect the growing importance of interdisciplinary research.

| Mixed Title | Title 1 | Title 2 |
|---|---|---|
| Bioconvection Transport Irradiation Suspensions: across Non-scattering Coarse-grain Molecular under Heating Fullerene in Membrane Collimated Above a Study Phototactic from Cell Dynamics of | Heating from Above in Non-scattering Suspensions: Phototactic Bioconvection under Collimated Irradiation | Coarse-grain Molecular Dynamics Study of Fullerene Transport across a Cell Membrane |
| Value of oil and gas Semi-intrusive exchange in the uncertainty quantity multiscale for stock gas london and change disclosures components oil of the relevance models propagation of upstream reserve companies | Value relevance of the components of oil and gas reserve quantity change disclosures of upstream oil and gas companies in the London Stock Exchange | Semi-intrusive uncertainty propagation for multiscale models |

Table 3: Overview of the dataset for the Mixed Title task.

The **Mixed Titles** task operationalizes this trend by blending two distinct paper titles into a single query. This challenges language models to identify relevant papers that address both topics, testing their ability to facilitate interdisciplinary exploration. The task evaluates not only the factual accuracy of model-generated references but also their capacity to support knowledge synthesis across domains.

A total of 265 mixed titles are generated by combining two randomly selected paper titles from the dataset. Table 3 shows sample mixed titles along with the original titles from which they were derived.

## 4 Methodology

In this section, we outline our evaluation pipeline designed to mimic realistic user interactions with language models. The pipeline is built around two tasks: the **Jumbled Titles** task and the **Mixed Titles** task.

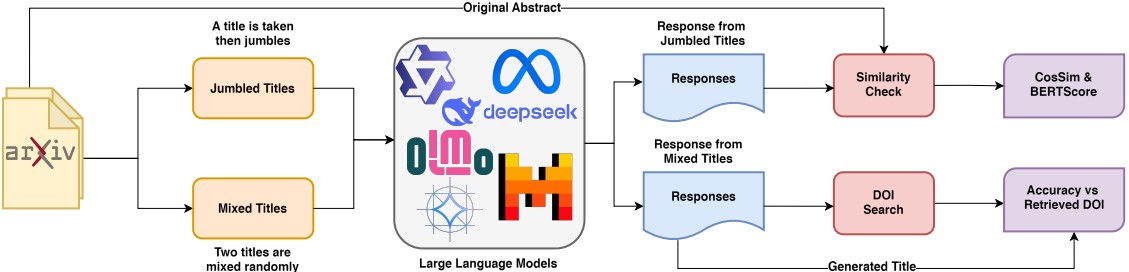

Figure 2: Pipeline for evaluating language models using the ArXiv dataset.

### 4.1 Jumbled Titles

For the Jumbled Titles task, we randomly select 5 titles per category from the ArXiv dataset and scramble the words within each title. The resulting jumbled titles serve as input prompts to the language models. The prompt template used is:

```
Tell me about this research paper: [jumbled title]
```

Here, *[jumbled title]* refers to the scrambled title. The language models' responses are evaluated by comparing them with the original abstracts. We generate embeddings using the *all-MiniLM-L6-v2* model Reimers & Gurevych (2019) and compute cosine similarity scores. In addition, BERTScore Zhang et al. (2020) and Semantic Textual Similarity (STS) Reimers & Gurevych (2019) metrics are employed to assess the response quality.

Algorithm 1 details the creation of the Jumbled Titles dataset.

---
**Algorithm 1** Create Jumbled Titles Dataset
---
**Require:** Parquet file path *parquet_file_path*, Output CSV file path *output_csv_file_path*
**Ensure:** CSV file with jumbled titles is saved
 1: **function** JUMBLE_TITLE(title)
 2:     Split the *title* into words
 3:     Randomly shuffle the words
 4:     **return** the shuffled words joined into a single string
 5: **end function**
 6: **procedure** CREATE_JUMBLED_TITLES_DATASET(parquet_file_path, output_csv_file_path)
 7:     Load the dataset from *parquet_file_path* into DataFrame *df*
 8:     Apply JUMBLE_TITLE(title) to the *title* column in *df*
 9:     Create a new DataFrame *jumbled_titles_df* with the jumbled titles
10:     Save *jumbled_titles_df* to CSV file at *output_csv_file_path* without the index
11: **end procedure**

---
**Algorithm 2** Create Mixed Titles Dataset
---
**Require:** List of titles *titles*
**Ensure:** List of mixed titles with original pairs
 1: **function** MIX_TITLES(title1, title2)
 2:     Split *title1* and *title2* into words
 3:     Concatenate the words from both titles into a list *mixed_words*
 4:     Randomly shuffle *mixed_words*
 5:     **return** the shuffled words joined into a single string
 6: **end function**
 7: **procedure** CREATE_MIXED_TITLES(titles)
 8:     Randomly shuffle the *titles* list
 9:     **if** the length of *titles* is odd **then**
10:         Append an empty string to *titles*
11:     **end if**
12:     Initialize an empty list *mixed_titles_data*
13:     **for** each pair of titles in *titles* **do**
14:         Mix the pair of titles using the *mix_titles* function
15:         Append a dictionary with keys *mixed_title*, *title1*, and *title2* to *mixed_titles_data*
16:     **end for**
17:     **return** *mixed_titles_data*
18: **end procedure**

---

## 4.2 Mixed Titles

For the Mixed Titles task, we generate mixed titles by combining two randomly selected paper titles from the dataset. This task is designed to emulate interdisciplinary queries where users combine concepts from different fields. The language models are prompted using the template:

```
Tell me 2 papers related to this and only mention the Title and the DOI: [mixed title]
```

Here, *[mixed title]* is the result of merging two titles. After generating responses, we evaluate the provided DOIs in two steps:

1. **DOI Validity Check:** Verify each model-generated DOI using API requests to databases such as Crossref, DataCite, UnPaywall, and OpenAlex.

2. **Title Accuracy Verification:** For validated DOIs, retrieve the official paper titles and compare them against the model-generated titles.

Algorithm 2 outlines the process for creating the Mixed Titles dataset.

## 5   Results

To run the inference on the models, we used 2× T4 Tesla GPUs (16GB each). Our experiments were conducted using PyTorch Paszke et al. (2019) and Huggingface's Transformers Wolf et al. (2020). The complete evaluation pipeline for each model required approximately 2.5 to 3 hours. To improve inference speed and efficiency, we applied 4-bit quantization using bitsandbytes bitsandbytes foundation (2024).

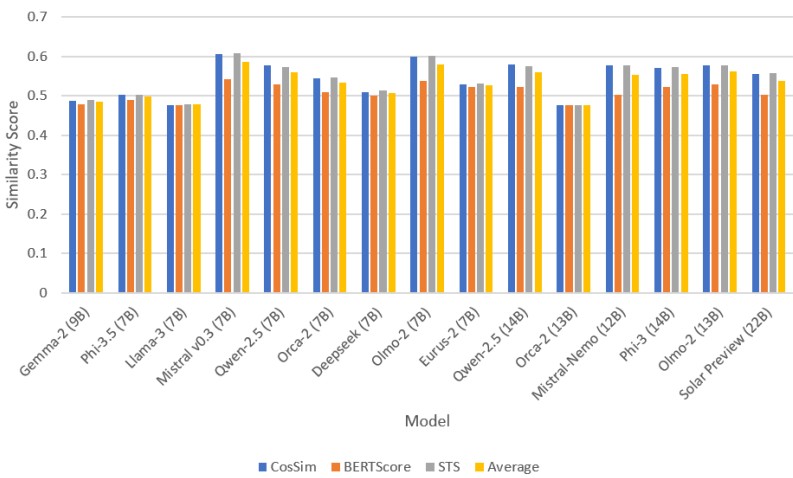

Figure 3: CosSim, BERTScore and STS Scores for all models.

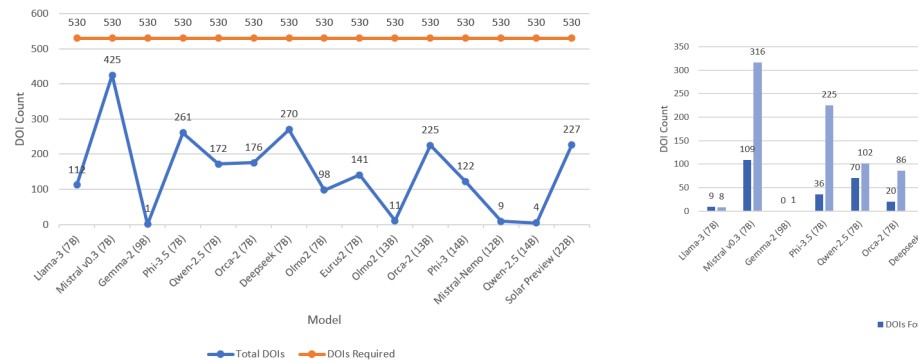

Figure 4: DOIs generated by each model during the *Mixed Title* task.

Figure 5: Comparison of DOIs Found vs. DOIs Not Found for each model.

Table 4 presents the performance of the models on the Jumbled Titles task. Notably, Mistral v0.3 Jiang et al. (2023) achieved the highest similarity scores, with a cosine similarity of 0.605, a BERTScore of 0.542, and an STS of 0.607. Qwen2.5 (7B) was the second best performing model overall. On average, BERTScore showed a 1.068% reduction in similarity compared to cosine similarity. The worst performing model was Orca-2 (13B), with scores of 0.476 (CosSim), 0.475 (BERTScore), and 0.477 (STS), averaging 0.476.

Table 5 evaluates the performance on the Mixed Titles task. Mistral-v0.3 (7B) generated the highest total number of DOIs (425), with 25.56% of them verified as valid. In contrast, Gemma-2 (9B) and Qwen-2.5 (14B)

| Model | Parameters | CosSim | BERTScore | STS | Average |
|---|---|---|---|---|---|
| Gemma-2 | 9B | 0.487 | 0.478 | 0.489 | 0.485 |
| Phi-3.5 | 7B | 0.502 | 0.489 | 0.503 | 0.498 |
| Llama-3 | 7B | 0.477 | 0.477 | 0.479 | 0.478 |
| Mistral v0.3 | 7B | **0.605** | **0.542** | **0.607** | **0.585** |
| Qwen-2.5 | 7B | 0.578 | 0.528 | 0.572 | 0.559 |
| Orca-2 | 7B | 0.544 | 0.508 | 0.546 | 0.533 |
| Deepseek | 7B | 0.509 | 0.501 | 0.513 | 0.507 |
| Olmo-2 | 7B | *0.600* | *0.537* | *0.602* | *0.580* |
| Eurus2 | 7B | 0.528 | 0.523 | 0.530 | 0.527 |
| Qwen-2.5 | 14B | 0.579 | 0.522 | 0.575 | 0.559 |
| Orca-2 | 13B | 0.476 | 0.475 | 0.477 | 0.476 |
| Mistral-Nemo | 12B | 0.577 | 0.503 | 0.578 | 0.553 |
| Phi-3 | 14B | 0.571 | 0.522 | 0.572 | 0.555 |
| Olmo-2 | 13B | 0.577 | 0.528 | 0.578 | 0.561 |
| Solar Preview | 22B | 0.55 | 0.502 | 0.558 | 0.537 |

Table 4: Cosine Similarity Scores, BERTScores and STS Scores between the generated response and the original abstract for various language models. Best performing model is shown in **Bold** and second best in *Italics*.

| Model | Total DOIs | DOIs Found | DOIs Not Found | Matching Titles |
|---|---|---|---|---|
| Llama-3 (7B) | 112 | 9 [8.04%] | 8 [91.96%] | 0.00% |
| Mistral v0.3 (7B) | 425 | 109 [25.65%] | 316 [74.35%] | 0.00% |
| Gemma-2 (9B) | 1 | 0 [0.00%] | 1 [100.00%] | 0.00% |
| Phi-3.5 (7B) | 261 | 36 [13.79%] | 225 [86.21%] | 0.00% |
| Qwen-2.5 (7B) | 172 | 70 [40.70%] | 102 [59.30%] | 0.00% |
| Orca-2 (7B) | 176 | 20 [18.87%] | 86 [81.13%] | 0.00% |
| Deepseek (7B) | 270 | 62 [22.96%] | 208 [77.04%] | 0.00% |
| Olmo2 (7B) | 98 | 12 [12.24%] | 86 [87.76%] | 0.00% |
| Eurus2 (7B) | 141 | 40 [28.37%] | 101 [71.63%] | 0.00% |
| Olmo2 (13B) | 11 | 4 [36.36%] | 7 [63.64%] | 0.00% |
| Orca-2 (13B) | 225 | 39 [17.33%] | 186 [82.67%] | 0.00% |
| Phi-3 (14B) | 122 | 35 [28.69%] | 87 [71.31%] | 0.00% |
| Mistral-Nemo (12B) | 9 | 2 [22.22%] | 7 [77.78%] | 0.00% |
| Qwen-2.5 (14B) | 4 | 0 [0.00%] | 4 [100.00%] | 0.00% |
| Solar Preview (22B) | 227 | 59 [25.99%] | 168 [74.01%] | 0.00% |

Table 5: DOI Search and Title Comparison Results for the Mixed Titles task.

generated 0% valid DOIs. The best DOI validity rate was achieved by Qwen-2.5 (7B) with 70 valid DOIs (40.70%). Figure 5 graphically depicts that for each model, the number of DOIs not found exceeds those found. Moreover, all models exhibited a consistent shortcoming: every valid DOI retrieved corresponded to an incorrect title (0% matching). It is noteworthy that Mistral-v0.3 (7B) generated the highest number of valid DOIs (109 out of 425).

We evaluate the factual consistency of various models using FactCCKryscinski et al. (2020), an entailment-based model designed to assess the accuracy of generated outputs for the Jumbled Titles Task. The results, as presented in Table 6, reveal that all models achieved high FactCC scores, ranging from 0.903 to 0.944. However, despite these high scores, every model was labeled as 'INCORRECT' for all 528 Jumbled Titles, meaning the model labeled it 'INCORRECT' with that much confidence. This discrepancy suggests that while the generated outputs may appear superficially similar to the expected results, they frequently contained factual inconsistencies or hallucinated information. This finding underscores the limitations of relying solely on surface-level similarity metrics for evaluating the factual accuracy of generated content.

| Model | Score | Label | Number |
|-------|-------|-------|--------|
| Solar (22B) | 0.943 | INCORRECT | 528 |
| Qwen2.5(14B) | 0.944 | INCORRECT | 528 |
| Qwen2.4(7B) | 0.937 | INCORRECT | 528 |
| Eurus2(7B) | 0.911 | INCORRECT | 528 |
| Phi-3(14B) | 0.936 | INCORRECT | 528 |
| Phi-3.5(7B) | 0.924 | INCORRECT | 528 |
| Orca(7B) | 0.935 | INCORRECT | 528 |
| Orca(13B) | 0.931 | INCORRECT | 528 |
| Olmo(13B) | 0.933 | INCORRECT | 528 |
| Olmo(7B) | 0.933 | INCORRECT | 528 |
| Mistral-Nemo(12B) | 0.909 | INCORRECT | 528 |
| Mistral v0.3 (7B) | 0.935 | INCORRECT | 528 |
| Llama-3(7B) | 0.913 | INCORRECT | 528 |
| Gemma-2(9B) | 0.903 | INCORRECT | 528 |
| deepseek (7B) | 0.936 | INCORRECT | 528 |

Table 6: FactCC scores and labels for the generated outputs of the models for the Jumbled Titles task.

## 6  Conclusion

This paper evaluates the extent of hallucination in state-of-the-art language models by designing two tasks: **Jumbled Titles** and **Mixed Titles**. In the Jumbled Titles task, the fifteen evaluated models achieved an average cosine similarity score of 0.544, 0.509 on BERTScore, and 0.545 on STS. Mistral-v0.3 was the best-performing model across all metrics, averaging 0.585 on the Jumbled Titles task and outperforming models twice and thrice its size.

For the Mixed Titles task, while models generated DOIs for the mixed titles, they often cited non-existent papers or mismatched DOIs. These results underscore critical limitations in maintaining factual accuracy in domain-specific contexts. On average, valid DOIs were generated only 17.75% of the time. Moreover, every model completely failed to generate the corresponding DOI for the title they generated, indicating that models struggle with maintaining factual consistency. To further highlight *Prompt Adherence*, it is worth noting that none of the models generated the required number of two DOIs for each *Mixed Title*. This discrepancy is evident as the expected output for each model was 530 DOIs (given 265 mixed titles), but none of the models met this requirement, as seen in Figure 4.

### 6.1  Model Size Performance

In Table 4 and Table 5, we observe a concerning trend where many of the larger models are significantly outperformed by their smaller counterparts in both tasks. For instance, the 7B Mistral-v0.3 outperforms models up to three times its size, while the Solar Preview (22B) demonstrates mediocre performance despite its substantially larger parameter count. A similar trend is seen with Qwen2.5, where the 7B variant outperforms the 14B variant. These findings raise serious questions about the relationship between model size and task performance. The results starkly highlight that simply scaling up model parameters does not guarantee superior performance in specialized tasks, particularly those requiring precise adherence to instructions and factual accuracy. This counterintuitive finding challenges the common assumption that larger language models inherently perform better, suggesting that architectural choices and training approaches might be more crucial than raw parameter count for achieving superior performance.

## 7  Limitations

There are several limitations to our work:

1. **Model Selection**: Our evaluation focuses on smaller models due to computational constraints. Results may differ significantly with larger variants, which could exhibit different performance characteristics. Although our findings in Section 6.1 shows that might not always be the case, as we observed smaller models often outperforming their larger counterparts.

2. **Model Quantization**: We use 4-bit quantization for inference. While this may reduce performance, studies suggest the impact is minimal Jin et al. (2024). The trade-off between computational efficiency and potential performance impact was deemed acceptable for our experimental setup.

3. **Human Evaluation**: Human evaluation remains a key limitation, as our current pipeline relies primarily on automated metrics like cosine similarity, BERTScore, and FactCC, which may not fully capture nuanced hallucinations or the scientific validity of generated outputs. Incorporating expert human assessments could provide deeper insights into relevance, factual correctness, and coherence, addressing gaps in purely quantitative evaluation.

4. **Data Contamination**: Given the scale of pretraining datasets, it is challenging to definitively determine whether specific papers in our test set were seen during model training. This makes it difficult to distinguish between genuine reasoning capabilities and potential memorization effects. Future work could address this by evaluating on recently published papers post-dating model training cutoffs, implementing systematic contamination checks, and using synthetic scientific papers to test reasoning capabilities.

## 8 Future Work

While ArXEval provides an automated pipeline for evaluating retrieval and generation in scientific language models, several avenues for future work can further enhance its scope and depth. A key direction is the integration of Retrieval-Augmented Generation (RAG) into the evaluation framework. Our current approach primarily assesses the intrinsic capabilities of language models in handling jumbled and mixed titles. RAG is a prominent technique for mitigating hallucinations by grounding language models in external knowledge sources. Future iterations of ArXEval could incorporate a RAG component, allowing us to evaluate how well models utilize retrieved scientific literature to generate factually accurate and contextually relevant outputs. This would necessitate expanding the evaluation to assess not only generation quality but also the effectiveness of the retrieval mechanism and the model's ability to faithfully incorporate retrieved information. Metrics specifically designed for RAG evaluations, such as retrieval precision/recall and faithfulness to the retrieved context, could be incorporated alongside our existing metrics.

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
