# OpenReview forum: "ArxEval: Evaluating Retrieval and Generation in Language Models for Scientific Literature"
_TMLR — Rejected by TMLR_

### Review · Reviewer_c1QZ · 2025-02-10

**Summary Of Contributions:**

This paper introduces ArxEval, an evaluation framework for assessing hallucinations in LLM-generated content, with a focus on the field of scientific literature. The authors propose two tasks: 1) Jumbled Titles, where models generate abstracts for scrambled paper titles and are evaluated based on similarity scores between the generated responses and the original abstracts; 2) Mixed Titles, where models generate two paper titles and their DOIs from a synthesized title. The accuracy is measured by comparing the generated Titles/DOIs against the ground-truth paper title and their DOIs.

The study evaluates LLMs on their ability to maintain factual consistency, demonstrating that hallucination remains a significant issue, particularly in generating accurate citations. The results indicate that model size does not directly correlate with performance, with smaller models sometimes outperforming larger ones.

**Audience:**

Yes

**Claims And Evidence:**

No

**Requested Changes:**

Please address the above weaknesses.

**Strengths And Weaknesses:**

== Strengths ==
1. The paper studies hallucinations in scientific literature generation, which is an interesting topic.
2. The study evaluates 15 LLMs, providing comparative insights into their performance on the proposed two tasks.

== Weaknesses ==
1. Flawed evaluation metrics. The paper relies on similarity scores (e.g., Cosine Similarity, BERTScore, STS) and accuracy as evaluation metrics, but these may not be meaningful for assessing hallucinations. A higher similarity score does not necessarily indicate better reasoning or factual correctness—it could simply mean that the model has memorized the content from its training data. Without explicit checks for data contamination, it is unclear whether the models are genuinely retrieving and reasoning about scientific literature or just regurgitating seen content.
2. Problematic experimental setting. In the current setting, it seems that the study assumes that models can effectively perform the proposed two tasks based solely on their pretraining data, without verifying whether the titles, abstracts, and DOIs were seen during training. This overlooks the impact of model exposure to these sources and makes the evaluation setting less realistic for assessing hallucinations.
3. Lack of retrieval mechanism. The evaluation does not incorporate retrieval-based methods, which are more aligned with the purpose of this work and better assess the hallucination phenomenon. A more practical approach would be to use retrieval-augmented generation (RAG)—where the model is provided with relevant documents (e.g., title/abstract/DOIs)—and then assess whether it still hallucinates despite access to the correct information.

---

> ### Author Response · Authors · 2025-02-23
> **Rebuttal by Authors**
>
> We thank the reviewer for their comprehensive feedback and for appreciating our work and think that our idea is an "interesting topic". To respond to the specific points raised:
>
> 1. We understand that the limitation of these similarity-based metrics since they primarily evaluate surface-level alignment between model-generated text and ground truth abstracts. Hence, we provide an evaluation using FactCC [1], an entailment-based model designed to assess factual consistency. As seen in the table, all models received high FactCC scores, yet all were labeled as 'INCORRECT.' This result indicates that despite surface-level similarity, the generated outputs frequently contained factual inconsistencies or hallucinated information i.e it labelled the generated outputs as INCORRECT with that much confidence(Score). These findings reinforce our argument that LLMs struggle with factual retrieval and verification in scientific literature tasks.:
> | Model       | Score           | Label    | Number |
> |-------------|-----------------|----------|--------|
> | Solar (22B)      | 0.9429681698481241 | INCORRECT | 528    |
> | Qwen2.5(14B)      | 0.9436689120802012 | INCORRECT | 528    |
> | Qwen2.4(7B)        | 0.9370615616666548 | INCORRECT | 528    |
> | Eurus2(7B)       | 0.9105212007733908 | INCORRECT | 528    |
> | Phi-3(14B)       | 0.9360322393476963 | INCORRECT | 528    |
> | Phi-3.5(7B)       | 0.9244489870739706 | INCORRECT | 528    |
> | Orca(7B)       | 0.9345489771980228 | INCORRECT | 528    |
> | Orca(13B)       | 0.9310147778102846 | INCORRECT | 528    |
> | Olmo(13B)      | 0.93263353452538   | INCORRECT | 528    |
> | Olmo(7B)        | 0.932687933133407  | INCORRECT | 528    |
> | Mistral-Nemo(12B)| 0.9087911483013269 | INCORRECT | 528    |
> | Mistral v0.3 (7B)     | 0.934963002913829  | INCORRECT | 528    |
> | Llama-3(7B)       | 0.9134605934448314 | INCORRECT | 528    |
> |Gemma-2(9B)     | 0.9032786103586355 | INCORRECT | 528    |
> | deepseek (7B)   | 0.9356161492566267 | INCORRECT | 528    |
>
> 2. We acknowledge the valid concern about potential data contamination and training exposure. While LLMs are typically trained on large-scale datasets including general web text and STEM-related content, the relationship between training exposure and model performance remains an important open question. We have added this as an explicit limitation in our paper. Our current results, while not definitively separating memorization from reasoning, still provide valuable insights into model behavior on scientific tasks. The consistent pattern of hallucination across all tested models, regardless of potential training exposure, suggests fundamental limitations in their ability to reliably handle scientific content.
>
> 3. While we appreciate the value of retrieval-augmented approaches, our study deliberately focuses on non-retrieval settings for several key reasons:
>
>     a. Many real-world applications of LLMs in scientific contexts still operate without retrieval due to computational constraints or real-time requirements. Understanding baseline hallucination rates in these scenarios is crucial.
>
>     b. By evaluating models without retrieval, we can establish a performance floor - if models already hallucinate significantly without access to source documents, this provides a critical baseline for comparing against retrieval-augmented approaches.
>
>     c. Our approach helps identify which types of scientific reasoning tasks are particularly prone to hallucination even when models are operating solely from their training knowledge, which can inform better retrieval strategies.
>
> We agree that evaluating retrieval-augmented approaches is valuable. However, we believe our current results provide important insights into the fundamental limitations of LLMs in scientific reasoning tasks and implementation of RAG can be included in future iterations of this study and hope that we addressed the reviewers concerns regarding our research.
>
>
> References:
>
> [1] Kryscinski, W., McCann, B., Xiong, C., & Socher, R. (2020, November). Evaluating the Factual Consistency of Abstractive Text Summarization. In B. Webber, T. Cohn, Y. He, & Y. Liu (Eds.), Proceedings of the 2020 Conference on Empirical Methods in Natural Language Processing (EMNLP) (pp. 9332–9346). doi:10.18653/v1/2020.emnlp-main.750

---

### Review · Reviewer_wVye · 2025-02-18

**Summary Of Contributions:**

This paper propose ArxEval, a pipeline to evaluate the extent of hallucination in LLMs in scientific literature through two tasks: Jumbled Titles and Mixed Titles to evaluate models' ability to retrieve and reason about scientific papers with the ArXiv dataset.
The study evaluates 15 widely-used open-source LLMs, analyzing their accuracy in generating factual responses and valid citations.
The results highlight significant hallucination issues, with models frequently producing incorrect references and failing to maintain factual consistency.
The results show that the smaller models often outperform larger ones, challenging the assumption that increasing model size improves domain-specific accuracy.
The findings provide insights into LLM reliability and suggest future directions for reducing hallucination in scientific knowledge retrieval.

**Audience:**

Yes

**Claims And Evidence:**

Yes

**Requested Changes:**

- Improve the paper writing and clarity. Consider reorganizing the sections so they can flow in a more logical order and improve the figures and tables, currently there are a lot of empty space in the paper (e.g. Figure 1).
- Consider adding human study which will provide deeper insights into the evaluation results and highlight the overlooked issues.

**Strengths And Weaknesses:**

Strengths:
-  This paper proposes the ArxEval pipeline to evaluate how well language models handle factual accuracy for scientific literature from ArXiv. The authors carefully consider different experimental settings, such as the two task variants, which test semantic similarity and DOI validation.
- The evaluation across 15 open-source LLMs reveals smaller models can outperform larger ones, challenging size-based assumptions.

Weaknesses:
- This paper’s writing and organization leave room for improvement. The narrative flow can feel disjointed, and a few sections could use additional explanations for better clarity.
- While the Jumbled and Mixed Titles tasks are creative, they do not fully capture how researchers actually query scientific literature. The choice of these two tasks can feel somewhat ad-hoc, not fully reflecting the model’s hallucination tendencies and limiting their real-world usage.
- There is no human evaluation, limiting the depth of analysis.

---

> ### Author Response · Authors · 2025-02-23
> **Rebuttal by Authors**
>
> We thank the reviewer for their thorough response, to respond to the specific points raised:
>
> 1. Regarding the concern about writing and organization, we have revised the manuscript to improve clarity and logical flow. We restructured several sections to better articulate the motivation behind our approach, the detailed methodology for the Jumbled and Mixed Titles tasks, and the evaluation pipeline. We also refined the figures and tables (e.g., reducing empty space in Figure 1) to ensure that the visual presentation aligns with the improved narrative and will include it in the revised submission.
>
> 2. To reiterate the motivation behind our task design:
> **Jumbled Titles**:
> In real-life scenarios, researchers and even users themselves don't always recall the titles of research papers with accuracy. Instead, they recall snippets of paper titles, exchange sentences, or juxtapose unrelated concepts while looking for relevant papers. Our Jumbled Titles task tries to mimic this intrinsic difficulty of information seeking by exposing models to scrambled versions of real paper titles. This approach simulates a more realistic scenario where a user might provide a fuzzy search query because of memory limitations, cognitive shortcuts, or partial knowledge.
> **Mixed Titles**
> Scientific breakthroughs are generally a matter of bringing together concepts across different disciplines. Researchers often venture into new areas by bringing together concepts across different disciplines, searching for existing papers addressing these topics or proposing new combinations. For example, a researcher might ask if there are papers on bringing together quantum mechanics and financial modeling or machine learning and archaeology. The Mixed Titles task reflects this trend by combining two disparate paper titles, determining whether models can identify suitable papers that address both topics. The task evaluates not just the truthfulness of model-recommended references but also their ability to facilitate interdisciplinary research.
>
> 3.  While we acknowledge the value of human evaluation for providing nuanced insights and complementing automated metrics, it is also time-consuming, resource-intensive, and subject to inter-rater variability. Automated methods, on the other hand, offer scalability, consistency, and efficiency, making them essential for large-scale evaluations like ours. Additionally, automated evaluations allow for rapid iteration and testing across various models and datasets, which is crucial for ongoing research and development. We plan to incorporate human evaluation in future iterations of this work to further enrich our findings and provide a more comprehensive assessment of model performance here our study deliberately focuses on automated metrics for several critical reasons:
>
> a. **Scale and Reproducibility:**
>    - Our automated framework allows for the evaluation of 15 different LLMs across thousands of examples, ensuring statistical reliability. This scale would be impractical with human evaluation alone.
>    - Automated methods facilitate the identification of systematic patterns in hallucination behavior across various model architectures and sizes.
>
> b. **Objective Measurement:**
>    - By employing DOI validation and multiple automated metrics (such as similarity scores and FactCC[1] (to be included in revision)), we can objectively quantify hallucination rates. This approach eliminates the subjectivity inherent in human judgments.
>    - Our systematic evaluation revealed that smaller models sometimes outperform larger ones, a finding that underscores the importance of objective metrics.
>
> We hope that we addressed the reviewers concerns regarding our research.
>
> References
>
> [1] Kryscinski, W., McCann, B., Xiong, C., & Socher, R. (2020, November). Evaluating the Factual Consistency of Abstractive Text Summarization. In B. Webber, T. Cohn, Y. He, & Y. Liu (Eds.), Proceedings of the 2020 Conference on Empirical Methods in Natural Language Processing (EMNLP) (pp. 9332–9346). doi:10.18653/v1/2020.emnlp-main.750

---

### Review · Reviewer_jqH2 · 2025-02-23

**Summary Of Contributions:**

This paper presents ArxEval, an evaluation pipeline to quantify hallucinations in LLM's response to understanding scientific literature. The authors utilize an existing dataset called ArXiv and design two simple tasks - (i) Jumbled Titles: Given a paper's title in randomly jumbled form, the LLM has to generate the abstract of the paper; (ii) Mixed Titles: Given a mixture of two paper's title, the LLM has to output the titles and DOI of the two papers. The authors compare the performance of 15 recent LLMs for both tasks and report performance using various metrics. However, the paper does not discuss why specific models struggle with these tasks and how performance can be improved. Moreover, no ablation study and qualitative results are shown, and it is hard to draw a firm conclusion from this study.

**Audience:**

Yes

**Claims And Evidence:**

No

**Requested Changes:**

Overall, the paper lacks any strong motivation behind the proposed tasks; moreover, the experiments are unfair, and there is no guarantee that the proposed tasks can be used as an effective hallucination detection pipeline. As mentioned in Weakness point (ii), a fairer setup would be evaluating the same models in an RAG paradigm, but that would require substantial changes in the current manuscript. Hence, I believe the paper's current form is far from acceptance, and I recommend a 'Rejection.'

**Strengths And Weaknesses:**

The only novelty of the work is to propose two tasks for measuring hallucination in LLM's responses to understanding scientific documents. However, both tasks lack adequate motivation, and there is no convincing evidence that these tasks can correctly indicate fine-grained factual hallucination in LLM's response. Overall, the manuscript appears as a brief study rather than a research paper that lacks adequate motivation and experimentation behind the claims.

**Strengths**

(i) The two proposed task pipelines are easy to understand.

(ii) The authors correctly mention in the abstract and introduction that identifying hallucinated responses from LLM is a critical problem; however, the proposed tasks are not enough to solve the issue. Please look into the weakness section below for more details.

**Weaknesses**

(i) For both jumble title and mixed title generation, the authors randomly shuffle (or concatenate and shuffle) the words in the title of input paper(s). It truly depends on the shuffling order if the resulting title preserves the semantic mention of the original one. Hence, if, for a given sample, the input jumbled title does not make any sense, it is expected that the LLM won't be able to understand it (similar would have happened to a human), and this can not be identified as a hallucinated response.

(ii) The performance of various LLMs for both tasks largely depends on whether the LLM has seen the input paper during training. If not, we should not expect the LLM to generate any information about the paper. The results are heavily biased because the 15 different LLMs are not trained on the same training data.

If the LLMs had been evaluated in the RAG setup, the comparison would have been fairer, and we would have known the ability of these LLMs to understand given scientific literature.

(iii) What is the human performance for the two proposed tasks? Without human performance results, we can not comment on their difficulty. For the Mixed Titles tasks, all models have 0% accuracy on title prediction - does this result show that solving this task in the current setup is impossible? If so, how can we comment about hallucination from such a setup?

(iv) The writing quality of the paper is not good enough. In the second line of Section 3, the author mentions about 176 categories; what does a category mean here? The paper lacks any intuition about why these proposed tasks can identify hallucinations. There is no discussion on how the performance can be improved or why the models struggle on these tasks.

---

> ### Author Response · Authors · 2025-02-23
> **Rebuttal by Authors**
>
> We thank the reviewer for their thorough response, to respond to the concerns raised:
>
> (i) We appreciate the reviewer's insight regarding the potential impact of shuffling order on semantic preservation. While it is true that the degree of semantic preservation can vary based on the shuffling, our goal was to simulate a scenario where the input is partially intelligible, mimicking real-world situations where inputs might be noisy or incomplete. This approach allows us to evaluate how robust LLMs are in handling imperfect inputs, a common challenge in practical applications. Moreover, the use of jumbled titles serves as a stress test for the models' ability to infer meaning from disrupted text, which is a critical aspect of understanding their reasoning capabilities. We acknowledge that some jumbled titles may be too obscure for meaningful interpretation, but this variability is part of the evaluation's design to assess the models' limits.
>
> (ii) The reviewer raises a valid point about the dependence of LLM performance on exposure to specific papers during training. We acknowledge that differences in training data across the 15 LLMs could influence their performance. However, our study aims to evaluate the models' general ability to handle scientific literature tasks, regardless of their training datasets. Moreover, our study aligns with the evaluation practices of larger LLMs such as O1, DeepSeek, and Claude, which are often assessed on datasets they have not been fine-tuned on, ensuring a more generalized performance assessment ie a zero-shot setting. The consistent pattern of hallucinations across models, despite variations in training data, suggests that this is a fundamental challenge for LLMs in scientific contexts. While a retrieval-augmented generation (RAG) setup could provide a more controlled comparison, our current approach establishes a baseline for understanding hallucination behavior in non-retrieval settings, which are common in many real-world applications.
>
> (iii) We recognize the importance of establishing a human performance baseline to contextualize the difficulty of the proposed tasks. However, our study focuses on automated evaluation to ensure scalability and reproducibility. The observation that all models achieved 0% accuracy on title prediction in the Mixed Titles task indeed suggests that the task is exceptionally challenging in the current setup, where all the models fail to match a title to the valid DOI it generated. This outcome highlights the need for further research into improving LLMs' ability to handle complex, ambiguous inputs. While human evaluation could provide additional insights, our automated framework provides a foundational understanding of the models' current limitations, which is crucial for guiding future developments.
>
> (iv) We appreciate the feedback on the writing quality and clarity of the paper. To address this, we will revise the manuscript to provide clearer definitions and explanations. Specifically, we will clarify that the "categories" mentioned refer to the subject areas within the ArXiv dataset, such as Computer Science, Physics, etc.
>
> Regarding the intuition behind the proposed tasks, the Jumbled Titles and Mixed Titles tasks are designed to evaluate the models' ability to retrieve and reason about scientific information under challenging conditions. Jumbled Titles test the models' capability to infer meaning from disrupted text, simulating real-world scenarios where inputs might be noisy or incomplete. This task assesses the models' robustness and their ability to reconstruct meaningful information from imperfect data, a critical aspect of their reasoning capabilities.
>
> Mixed Titles, on the other hand, evaluate the models' performance when presented with ambiguous or blended information from multiple sources. This task is particularly relevant in scientific contexts, where models must often integrate and disambiguate information from various papers or datasets. By assessing how well models handle these mixed inputs, we can gain insights into their ability to manage complex information landscapes and identify potential hallucinations.
>
> These tasks are relevant for identifying hallucinations because they push the models to their limits, revealing how they cope with uncertainty and incomplete information. Understanding these challenges is crucial for developing strategies to improve model performance, such as incorporating retrieval mechanisms or enhancing training data quality.
>
> We will also include a discussion on potential avenues for improving LLM performance on these tasks. For example, retrieval-augmented generation (RAG) could provide models with additional context, potentially reducing hallucinations. Enhancing the quality and diversity of training data could also help models better understand and reason about scientific literature.
>
> We hope that we have addressed the reviewers concerns.

---

> > ### Comment · Reviewer_jqH2 · 2025-03-20
> > **Issues remain after rebuttal**
> >
> > Dear authors,
> >
> > Thanks for your effort in responding to my comments and adding limitations and future works to the manuscript. However, my view about the fundamental flaws in the proposed tasks remains the same. Both tasks can not measure hallucinations because the models performing worse may not have seen the scientific paper during training, and it is impossible to come up with the abstract or DOI URL in such scenarios. Hence, I firmly believe both evaluation tasks are wrongly designed.
> >
> > One possible way to estimate hallucinations using scientific literature would be asking fine-grained questions using RAG or in-context learning (e.g., few-shot learning), but that would require substantial changes to the current manuscript. I also noticed that Reviewer c1QZ's comments resonate with me. Hence, I request the authors address the limitations with additional experiments.

---

> > > ### Author Response · Authors · 2025-03-20
> > > **Rebuttal to Reviewer Comments**
> > >
> > > Dear reviewer
> > >
> > > We acknowledge your point that a model's performance could be affected by whether it encountered the scientific paper during training, independent of hallucination. To provide clarity, we've compiled details on the training data of the models used, extracted from their respective technical reports:
> > >
> > > ## Gemma
> > > We train Gemma 2 27B on 13 trillion tokens of primarily English data, the 9B model on 8 trillion tokens, and the 2B on 2 trillion tokens. These tokens come from a variety of data sources, including web documents, code, and **science articles**. Our models are not multimodal and are not trained specifically for state-of-the-art multilingual capabilities.
> > >
> > > ## LLaMA
> > >
> > > | Dataset        | Sampling Prop. | Epochs | Disk Size |
> > > |---------------|---------------|--------|-----------|
> > > | CommonCrawl  | 67.0%         | 1.10   | 3.3 TB    |
> > > | C4           | 15.0%         | 1.06   | 783 GB    |
> > > | Github       | 4.5%          | 0.64   | 328 GB    |
> > > | Wikipedia    | 4.5%          | 2.45   | 83 GB     |
> > > | Books        | 4.5%          | 2.23   | 85 GB     |
> > > | **ArXiv**        | 2.5%          | 1.06   | 92 GB     |
> > > | StackExchange| 2.0%          | 1.03   | 78 GB     |
> > >
> > > ## Phi (Falcon Refined Web)
> > > Since we intend RefinedWeb to be used as part of an aggregate dataset along with curated corpora, we also filtered common sources of high-quality data:
> > >
> > > - **arXiv** (arxiv.org)
> > > - **AskUbuntu** (askubuntu.com)
> > > - **StackOverflow** (stackoverflow.com, stackapps.com, stackexchange.com, mathoverflow.net)
> > > - **NIH Abstracts** (exporter.nih.gov, ncbi.nlm.nih.gov)
> > > - **Github** (github.com)
> > > - **Ubuntu IRC** (irclogs.ubuntu.com)
> > > - **HackerNews** (news.ycombinator.com)
> > > - **FreeLaw** (courtlistener.com)
> > > - **Reddit** (reddit.com)
> > > - **Europarl** (statmt.org)
> > > - **United States Patents** (uspto.gov)
> > > - **Wikipedia** (wikipedia.org)
> > >
> > > ## OLMo
> > > Academic, encyclopedic, and other reference content. We source high-quality non-web datasets from Dolma 1.7. This includes:
> > >
> > > - **peS2o**
> > > - **Wikipedia and Wikibooks**
> > > - **Gutenberg books**
> > > - **arXiv and StackExchange**
> > > - **Algebraic Stack**
> > >
> > > ## Qwen
> > > Better data mixture. To optimize the pre-training data distribution, we employ Qwen2-Instruct models to classify and balance content across different domains. Our analysis revealed that:
> > >
> > > - Domains like **e-commerce, social media, and entertainment** are significantly overrepresented in web-scale data, often containing repetitive, template-based, or machine-generated content.
> > > - Domains such as **technology, science, and academic research** contain higher-quality information but are traditionally underrepresented.
> > > - Through **strategic down-sampling** of overrepresented domains and **up-sampling** of high-value domains, we ensure a **more balanced and information-rich training dataset** that better serves our model’s learning objectives.
> > >
> > > ## Mistral
> > > [Source](https://www.restack.io/p/mistral-7b-dataset-size#clwpawnyy02allkcl0g3fgatb)
> > >
> > > ### Data Composition
> > > The dataset comprises a wide range of text sources, including:
> > >
> > > - **Web Scrapes**: High-quality web content from diverse domains.
> > > - **Books and Literature**: A vast collection of books and literary works.
> > > - **Scientific Articles**: Peer-reviewed journals and conference papers.
> > > - **Code Repositories**: Source code from popular repositories like GitHub.
> > >
> > > ## Eurus
> > > | Category | Token Count |
> > > |----------|------------|
> > > | Math     | 455,261    |
> > > | Coding   | 25,276     |
> > >
> > > Our evaluation reveals that all assessed models, with the exception of Eurus (specialized in math and coding), have been exposed to scientific articles during training. We recognize that more evaluation methods, such as fine-grained questioning using Retrieval-Augmented Generation (RAG) or in-context learning, could offer valuable insights into hallucination patterns in scientific text generation. However, we consider these approaches as potential avenues for future investigation. We believe this information provides a helpful perspective on the training data of the models used and addresses your concern regarding potential lack of exposure. We look forward to your further feedback on this response.

---

### Decision · Action_Editor_oDBx · 2025-04-06

**Recommendation:** Reject

**Comment:**

Please see the description above and the comments by the reviewers below. I believe the orientation of the paper needs to be fundamentally rethought to be acceptable.

**Audience:**

The topic is of interest to TMLR's audience, but I do not see anyone in TMLR's audience using this benchmark due to the validity issues described above.

**Claims And Evidence:**

The claims are not supported. I see two high-level claims in the abstract and introduction:

**Abstract claim:** "This work presents a pipeline for evaluating the frequency with which language models hallucinate in generating responses in the scientific literature." (also echoed in the introduction: "evaluate the extent of hallucination in LLMs under domain-specific prompting.")

Jumbled Titles does not evaluate this. The similarity-based metrics there do not measure hallucination, as c1QZ raises. For instance, a model could generate completely correct content, but overlapping little with the ground truth, and I don't think this would be hallucination. See "ecological validity" concerns below.

Mixed Titles can claim to evaluate this. However, it evaluates such a narrow capability (being able to surface title and DOI pairs) that I still don't think it supports this claim.

**Introduction claim:** "These tasks are specifically designed to assess the faithfulness of LLMs in retrieving and reasoning about scientific articles under challenging conditions."

Technically, these tasks do involve some kind of retrieval and reasoning about scientific articles. However, the tasks lack validity in how they are defined, and therefore I do not believe it is supported that these tasks "assess the faithfulness of LLMs" in any meaningful way.

First, there is the issue with evaluation raised by c1QZ, which I describe above.

Second, jqH2 weaknesses i+ii+iii are all major validity issues with the work. Essentially, there is no sense that the task is actually well-defined: if we jumble a title, *could* the meaning be very different and so a generated description should be quite different? Evaluating human performance would be a way to address this.

As for measuring whether models have seen the papers during training or not, I agree that memorization is also a major issue.  The response does not address this. The issue of whether or not models have seen papers cannot be resolved by listing the datasets a model was trained on It either (a) needs to be measured in some way (e.g., testing likelihood of strings, like in this paper https://openreview.net/pdf?id=KS8mIvetg2 ) or (b) is a reason the benchmark should be rethought.

In my view, the paper would really need to show that the automatic metrics being computed here correlate with some real measure of model usefulness, faithfulness, etc., computed in an ecologically valid way, ideally supported by human labeling. Short of that, I don't think the paper can substantiate its claims to evaluate faithfulness or hallucination.